# A taxonomic and molecular survey of the pteridophytes of the Nectandra Cloud Forest Reserve, Costa Rica

**Joel H. Nitta** [1]\*, **Atsushi Ebihara**[2], **Alan R. Smith**[3]

1 Department of Biological Sciences, Graduate School of Science, The University of Tokyo, Bunkyo-ku, Tokyo, Japan, 2 Department of Botany, National Museum of Nature and Science, Tsukuba, Ibaraki, Japan, 3 The University Herbarium, University of California at Berkeley, Berkeley, California, United States of America

\* joelnitta@gmail.com

## Abstract

Floristic surveys are crucial to the conservation of biodiversity, but the vast majority of such surveys are limited to listing species names, and few take into account the evolutionary history of species. Here, we combine classical taxonomic and molecular phylogenetic (DNA barcoding) approaches to catalog the biodiversity of pteridophytes (ferns and lycophytes) of the Nectandra Cloud Forest Reserve, Costa Rica. Surveys were carried out over three field seasons (2008, 2011, and 2013), resulting in 176 species representing 69 genera and 22 families of pteridophytes. Our literature survey of protected areas in Costa Rica shows that Nectandra has an exceptionally diverse pteridophyte flora for its size. Plastid *rbcL* was selected as a DNA barcode marker and obtained for >95% of pteridophyte taxa at this site. Combined molecular and morphological analyses revealed two previously undescribed taxa that appear to be of hybrid origin. The utility of *rbcL* for species identification was assessed by calculating minimum interspecific distances and found to have a failure rate of 18%. Finally we compared the distribution of minimum interspecific *rbcL* distances with two other areas that have been the focus of pteridophyte molecular surveys: Japan and Tahiti. The comparison shows that Nectandra is more similar to Japan than Tahiti, which may reflect the biogeographic history of these floras.

## Introduction

Despite its small area (51,100 km$^2$), Costa Rica is home to remarkably high biodiversity, and is ranked as one of the world's top 25 biodiversity hotspots [1]. It is estimated that vascular plant species richness in Costa Rica exceeds 5,000 spp. per 10,000 km$^2$ [2]. Accordingly, primary taxonomy is an important part of biological research in Costa Rica for the purposes of both documenting this biodiversity and informing conservation practices.

In addition to taxonomic diversity, measures of phylogenetic diversity (PD) provide an important perspective on biodiversity and should be taken into consideration for setting conservation priorities [3]. Therefore, surveys documenting both species richness and molecular phylogenetic diversity are needed. Furthermore, the use of a standard molecular marker as a

**Data Availability Statement:** Code to replicate all analyses, figures, and this manuscript is available at https://github.com/joelnitta/nectandra_ferns. A Docker image is available to run the code at https://hub.docker.com/r/joelnitta/nectandra_ferns. Data

are available from the Dryad repository at https://doi.org/10.5061/dryad.bnzs7h477. Newly generated sequences are deposited in GenBank (accession numbers MW138110-MW138295). Photos of voucher specimens available at https://www.fernsoftheworld.com/florulas/nectandra.

**Funding:** Field work and publication fees were funded in part by the Nectandra Institute, which was consulted and approved the decision to publish but had no other role in study design, data collection and analysis, or preparation of the manuscript. Molecular work was funded by a Japan Society for the Promotion of Science (JSPS) Kakenhi grant to AE (15K07204). Publication fees were funded in part by a grant to JHN from the Alan R. Smith Fern Research and Curation Fund. Neither JSPS nor the Alan R. Smith Fern Research and Curation Fund had any role in study design, data collection and analysis, decision to publish, or preparation of the manuscript.

**Competing interests:** The authors have declared that no competing interests exist.

DNA "barcode" is useful for species identification and taxonomic revision [4]. For example, DNA barcoding surveys have revealed hidden biodiversity in the form of cryptic species in butterflies in Costa Rica [5] and have been used to identify cryptic life-cycle stages in ferns [6,7]. DNA barcodes are being increasingly integrated into biological surveys to document biodiversity at unprecedented scale, rate, and resolution [8–10].

Pteridophytes (i.e., ferns and lycophytes) are an important group of plants to study because many new species are still being discovered in the Neotropics [11] and they play important ecological roles in many ecosystems [12–15]. The pteridophytes of Costa Rica comprise *c.* 1,200 spp., accounting for one-quarter to one-third of the estimated richness of the Neotropics (*c.* 3,000 spp. to 4,500 spp.) [11]. While the pteridophyte floras of some sites in Costa Rica are well studied, such as La Selva Biological Station [16,17], other areas in the country have received considerably less attention. We present here the results of surveys conducted over three field seasons on the pteridophyte flora of the recently established Nectandra Cloud Forest Reserve near San Ramon, Costa Rica. We also characterize the phylogenetic diversity at this site and compare it with other pteridophyte floras that have recently been the focus of DNA barcoding surveys.

## Materials and methods

### Study site

The Nectandra Cloud Forest Reserve (hereafter, "Nectandra"; 10˚11' N, 84˚31' W) is located at 1,000 m to 1,200 m a.s.l. on the Atlantic slope of the Cordillera de Tilarán, Alajuela Province (Fig 1A). It encompasses 158 ha of premontane rainforest (life zones follow Holdridge [18]) and is managed by the Nectandra Institute, whose mission is to conserve and restore cloud forest in northern Costa Rica through grassroots outreach [19]. Approximately three-quarters of Nectandra is primary forest with >98% canopy cover; the remaining area comprises naturally regenerating former coffee and *Dracaena* plantations (Fig 1B and 1C). Two permanent streams and four seasonal drainages pass through the reserve and empty into the Balsa river. Most of the surrounding land is used for cattle pasture or other agriculture. Previous biological surveys of Nectandra indicate that it has a rich herpetofauna [20] and is home to at least 188 spp. of bryophytes [21].

Climate at Nectandra is characterized by extremely high frequency of cloud cover throughout the year. Rainfall peaks during the wet season from November to February. Mean annual precipitation is 3,000 mm yr$^{-1}$ to 3,500 mm yr$^{-1}$, with *c.* 80% fog-saturated days.

### Field survey

We carried out surveys of pteridophytes over three field seasons (January 2008, 2011, and 2013; 37 days of sampling total). Most specimens were collected along trails through the reserve. Epiphytes were collected from fallen trees or tree branches, or up to *c.* 2 m on tree trunks. Permits for collection were obtained from the Costa Rican government (SINAC No. 04941 and Cites 2014-CR 1006/SJ [#S 1045]). The first set of voucher specimens was deposited at UC, with duplicates at CR, GH, TI, and the private collection at Nectandra. Herbarium codes follow Thiers [25]. Leaf tissue was preserved on silica gel for DNA extraction. Spores of selected taxa were observed with a standard compound light microscope.

### Taxonomy

We consulted relevant floras [26–28] and recent monographs [29–35] for species identification. In some cases, we also consulted taxonomic experts on particular groups (e.g., grammitid ferns). Genus-level and higher taxonomy follows Pteridophyte Phylogeny Group I [36].

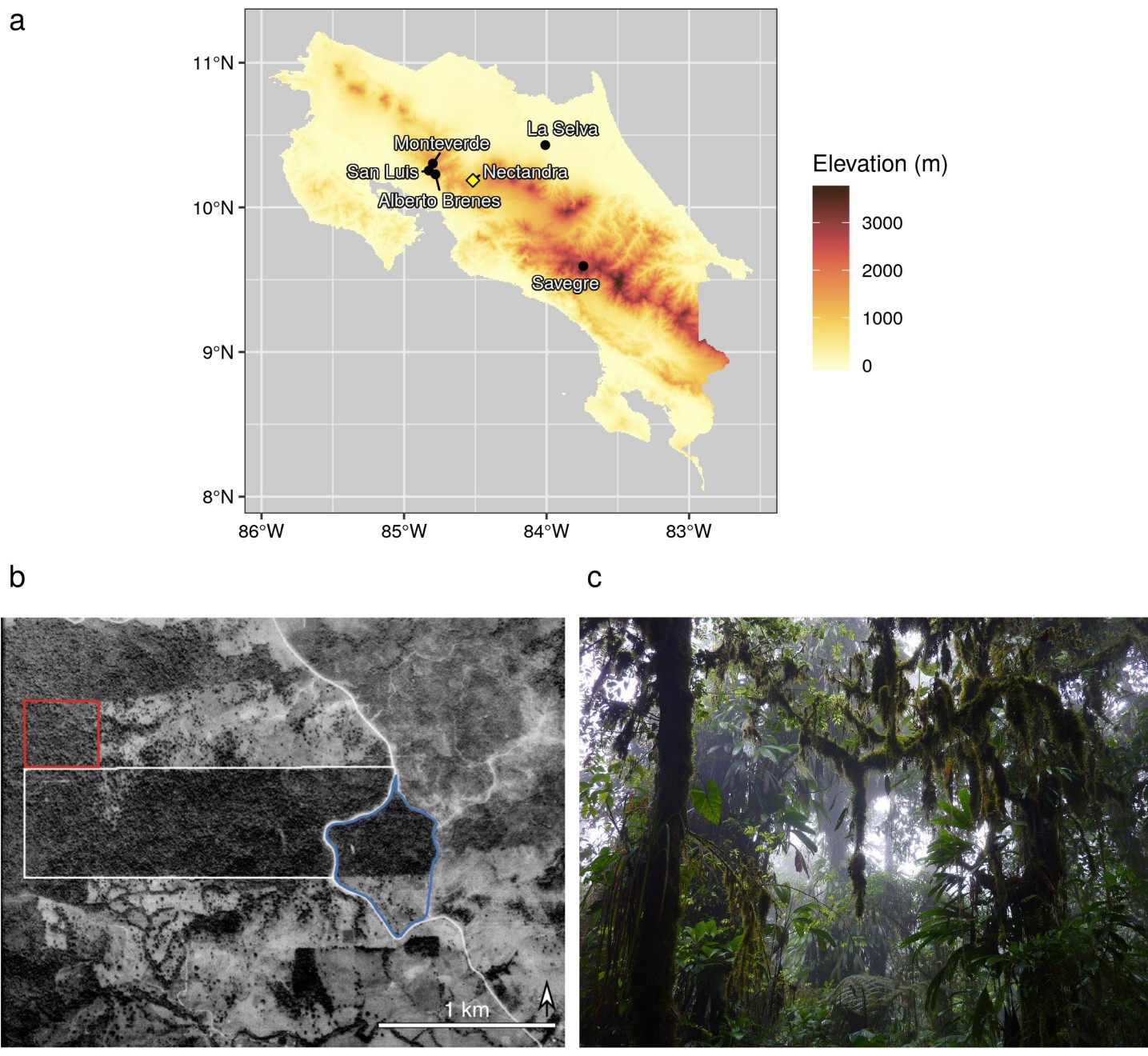

**Fig 1. Location of the Nectandra Cloud Forest Reserve ("Nectandra").** (a) Map of Costa Rica showing location of Nectandra (yellow diamond) and other protected areas that have been surveyed for pteridophytes (black dots). Other protected areas include Alberto Manuel Brenes Biological Reserve, La Selva Biological Station, Monteverde Cloud Forest Reserve, San Luis Biological Reserve, and the upper watershed of the Savegre River in the Los Santos Forest Reserve. Elevation data downloaded from the CGIAR SRTM 90m digital elevation database [22] using the getData function in the R package raster [23]. Map created using ggplot2 [24]. (b) Aerial photo of Nectandra (*c.* 1992). Nectandra consists of three parcels of land: The original parcel is in white (*c.* 80% primary forest), with areas added later in red ("Ocotea" parcel, primary forest) and blue ("Persea" parcel, secondary forest). Note extreme deforestation outside of protected areas. Reprinted under a CC BY license with permission from Instituto Geográfico Nacional de Costa Rica, original copyright 1992. (c) Example of interior of primary forest at Nectandra. Photo courtesy of Evelyne Lennette, reprinted under a CC BY license with permission.

## DNA sequencing and phylogenetic analysis

DNA was extracted with the DNEasy Plant Mini kit following the manufacturer's protocol (Qiagen). We generally sampled one species per taxon from Nectandra for morphologically distinct taxa, and up to five specimens per taxon for taxa that are more difficult to identify using standard keys [26,27] (e.g., *Megalastrum*, *Didymoglossum*). We selected the plastid *rbcL* gene as a barcode marker because it has universal primers available for ferns [37] and has performed relatively well for species identification in ferns relative to other candidate barcode loci [6,38,39]. We amplified *rbcL* using PCR primers and thermocycler settings of [37] and verified amplification success by gel electrophoresis in 1% TAE. We purified PCR products with Exo-STAR (GE Healthcare) and conducted cycle sequencing using the Big Dye Terminator v3.1 Cycle Sequencing Kit (ThermoFisher) with two internal primers, ESRBCL654R and ESRBCL628F [37], in addition to the amplification primers. We imported the resulting AB1 trace files into Geneious [40], assembled contigs, and exported the consensus sequences in FASTA format. We downloaded *rbcL* sequences from GenBank if available for any remaining taxa that could not be successfully sequenced (S1 Table). We generated an alignment using MAFFT [41]. For phylogenetic analysis by maximum likelihood, all sites were included in a single partition (no partitioning was specified). We evaluated models of DNA sequence evolution with IQ-TREE [42] ("-m TEST"), which is similar to the model selection process implemented in jModelTest [43]. The Bayesian Information Criterion (BIC) was used to select the best model (IQ-TREE default setting), which was then used by IQ-TREE to infer the tree. Node support was assessed with 1,000 ultra-fast bootstrap (UFboot) [42] and 1,000 Shimodaira-Hasegawa-like approximate likelihood ratio test (SH-aLRT) replicates [44] in IQ-TREE. For a small number of genera that were not supported as monophyletic in the original phylogenetic analysis, we also downloaded all available *rbcL* sequences for closely related taxa (at the family or subfamily level) from GenBank, aligned these in combination with the newly generated sequences from Nectandra with MAFFT, and inferred a tree using FastTree on default settings [45,46]. Molecular analysis was performed under permits R-CM-RN-001-2014-OT-CONAGEBIO and R-CM-RN-002-2017-OT-CONAGEBIO.

## Statistical analysis

To assess the completeness of sampling, we constructed species incidence rarefaction-extraction curves using the iNEXT package [47]. Number of collection days was used as the sampling unit.

We compared the number of species at Nectandra with other protected sites in Costa Rica by conducting a literature survey.

We assessed the utility of *rbcL* for species identification by calculating minimum interspecific distances as follows. We first calculated all raw interspecific distances in the *rbcL* alignment using the "dist.dna" function in the APE package [48], then extracted the minimum interspecific distance for each species using custom scripts. Species sharing identical *rbcL* sequences with at least one other species (interspecific distance of zero) were considered failures, i.e., not possible to identify with this marker.

To better characterize observed phylogenetic diversity, we compared the phylogenetic diversity of the pteridophytes of Nectandra with two other pteridophyte floras that have been the subject of DNA barcoding using *rbcL*: Japan [39,49] and the islands of Moorea and Tahiti, French Polynesia [6]. To ensure that phylogenetic distances were comparable across datasets, we generated a combined *rbcL* alignment for all species from the three floras together using MAFFT. We then subset the alignment to the species in each flora and calculated minimum interspecific distances per flora as described above.

**Table 1. Growth habits of ferns and lycophytes at the Nectandra Cloud Forest Reserve, Costa Rica.** Count includes taxa at the species and infraspecies (variety or subspecies) levels (*n* = 178 total).

|  | *n* (percent of total[a]) |
| --- | --- |
| Climbing or clambering | 5 (2.8%) |
| Epipetric | 19 (10.7%) |
| Epiphytic | 94 (52.8%) |
| Terrestrial | 74 (41.6%) |

[a]Percentages do not sum to 100% because some taxa have multiple growth habits.

All analyses were carried out using R v 3.6.1 [50].

## Results

### Taxonomic survey

Our surveys resulted in 320 individuals representing 176 spp., 69 genera, and 22 families of pteridophytes (S2 Table). Two species included multiple varieties (two each). 169 spp. (94.9%) are ferns and 7 spp. (3.9%) are lycophytes. Most taxa are either epiphytic (*n* = 94; 52.8%) or terrestrial (*n* = 74; 41.6%) (Table 1). All taxa are native, except for *Macrothelypteris torresiana* (Gaudich.) Ching (native to Africa and Asia; introduced in the Americas), which was excluded from the comparison of richness across sites and DNA barcode analysis.

The genera with the most species were *Elaphoglossum* (14 spp.), *Diplazium* (12 spp.), *Hymenophyllum* (12 spp.), and *Asplenium* (10 spp.). Families with the most species were Dryopteridaceae (28 spp.), Polypodiaceae (28 spp.), and Hymenophyllaceae (25 spp.).

Three taxa could not be matched to any known species: *Polyphlebium* sp1 (*Nitta 123* and *Nitta 2378*), *Campyloneurum* sp1 (*Nitta 2308*), and *Megalastrum* sp1 (*Nitta 727*).

Our literature survey identified five other protected areas in mainland Costa Rica that have been surveyed for pteridophytes [17,52–55] (Fig 1, Table 2). The site with the highest species richness is Alberto Manuel Brenes Biological Reserve (281 spp.), with Nectandra (175 spp.) the third-highest after La Selva Biological Station (197 spp.; Table 2). Nectandra has by far the most species per hectare (1.11), with San Luis Biological Reserve second-highest (0.16 species per hectare; Table 2).

The collection curve did not reach an asymptote (Fig 2). Extrapolation of the curve indicates that asymptotic species richness may approach 253 spp. (95% confidence interval 222 spp. to 305 spp.).

**Table 2. Species richness of pteridophytes at protected areas in Costa Rica.** A single non-native species, *Macrothelypteris torresiana*, was excluded from calculations for Nectandra.

|  | Elevation (m) | Life zone type[a] | Area (ha) | Richness (no. spp.) | Richness per area (no. spp. per ha) | Reference |
| --- | --- | --- | --- | --- | --- | --- |
| Alberto Manuel Brenes Biological Reserve | ca. 1,520 | Premontane rainforest | 7,800 | 281 | 0.04 | [54] |
| La Selva Biological Station | 35–130 | Tropical wet forest | 1,533 | 197 | 0.13 | [17] |
| Monteverde Cloud Forest Reserve | 1,500–1,640 | Lower montane wet forest | 3,800 | 147 | 0.04 | [52] |
| Nectandra Cloud Forest Reserve | 1,000–1,200 | Premontane rainforest | 158 | 175 | 1.11 | this study |
| San Luis Biological Reserve | 540–855 | Transition between lowland tropical wet forest and premontane wet forest | 251 | 39 | 0.16 | [55] |
| Upper watershed of the Savegre River in the Los Santos Forest Reserve | 2,000–3,491 | Montane forest to paramo | 10,000 | 123 | 0.01 | [53] |

[a]Life zones follow Holdridge [18].

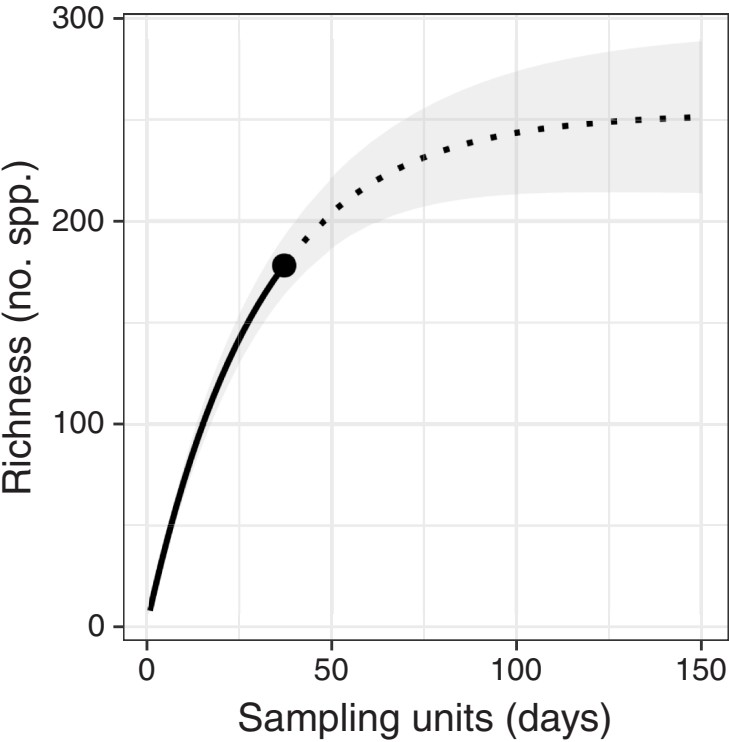

**Fig 2. Interpolation (solid line) and extrapolation (dashed line) of species richness of pteridophytes at Nectandra Cloud Forest Reserve, Costa Rica.** Point at transition from solid to dashed line indicates observed richness. Gray shading indicates 95% confidence interval.

## Phylogenetic analysis

We generated 186 new *rbcL* sequences of pteridophytes from Nectandra, representing 168 taxa (S1 Table). Mean length of the newly generated sequences was 1,292 bp ± SD 96 bp. *rbcL* could not be sequenced for 10 taxa, and only partial sequences (ca. 3' or 5' half of *rbcL*) could be obtained for three taxa. Difficulty in sequencing of these samples may be due to degraded condition of DNA or mismatched primer sequences. Of the taxa that we were unable to newly sequence, sequences of two taxa from specimens from Nectandra and three taxa from specimens from other areas were available on GenBank (S1 Table). In total, our *rbcL* sampling included 173 taxa (97.2%) of pteridophytes occurring at Nectandra.

The *rbcL* alignment was 1,309 bp long including 591 parsimony-informative sites and 191 sequences. The GTR+I+G4 model was selected for ML phylogenetic analysis according to BIC (the same model was also selected by AIC and corrected AIC).

The *rbcL* phylogeny (S1 Fig) was generally in agreement with recently published plastid phylogenies [37,56–58] at the family level and above, and families *sensu* PPGI [36] were monophyletic. One exception was Osmundaceae sister to Gleicheniaceae (SH-aLRT <50%; UFboot 92%), which may be due to poor sampling; we could obtain only a 614 bp *rbcL* fragment from the sole osmundaceous species at our study site. Some internal nodes had weak support (e.g., Cyatheales [SH-aLRT 64.7%; UFboot 92%], Polypodiidae [leptosporangiates; SH-aLRT <50%; UFboot 87%]). It should be noted that IQ-TREE UFboot support values are not analogous to traditional bootstrap values; only nodes receiving SH-aLRT > = 80% and UFboot > = 95% should be considered reliable [59]. As the purpose of this study was not to robustly infer phylogeny across all ferns and lycophytes, we do not discuss such deep relationships further.

A small number of genera were found to be non-monophyletic but lacked strong support. *Sphaeropteris brunei* (H. Christ) R.M. Tryon is nested within *Cyathea* as the sister to *Cyathea bicrenata* Liebm. (SH-aLRT 79.6%; UFboot 83%). *Stenogrammitis limula* (Christ) Labiak is nested within *Lellingeria* as the sister to *Lellingeria hombersleyi* (Maxon) A.R. Sm. (SH-aLRT <50%; UFboot 87%). These were resolved in their expected genera in broadly sampled trees using all available *rbcL* sequences on GenBank (S2 Fig and S3 Fig, respectively), so their irregular placement in the Nectandra *rbcL* tree appears to be an artifact of low sampling rather than poor sequence quality or misidentification.

Sampling of multiple specimens per species for taxa that are morphologically difficult to distinguish revealed several non-monophyletic species (S1 Fig). *Megalastrum apicale* R.C. Moran & J. Prado, *M. atrogriseum* (C. Chr.) A.R. Sm. & R.C. Moran, and *M. longipilosum* A. Rojas are closely related (SH-aLRT 92%; UFboot 100%), and the monophyly of each lacks support. *Diplazium carnosum* Christ is non-monophyletic with respect to *D. urticifolium* Christ and *D. macrophyllum* Desv., but this and most other relationships within *Diplazium* lacked support. *Didymoglossum ekmanii* (Wess. Boer) Ebihara & Dubuisson is nested within, and very closely related to *D. kapplerianum* (J.W. Sturm) Ebihara & Dubuisson (SH-aLRT 100%; UFboot 100%).

## Barcode analysis

The pteridophyte flora of Nectandra includes 31 taxa (18%) that share identical *rbcL* sequences with at least one other taxon. This failure rate is higher than that of the pteridophytes of Moorea and Tahiti (4%), but lower than Japan (22%; Fig 3).

## Discussion

Here, we present to our knowledge the first combined taxonomic and molecular survey of pteridophytes of a protected area in Costa Rica. We place our results in a regional and global context by comparing this flora with other protected areas in Costa Rica and two other sites that have been the focus of DNA barcoding: Tahiti and Japan.

### Taxonomic diversity

Nectandra has the third-highest species richness of protected areas in Costa Rica with data available for ferns and lycophytes, and by far the greatest number of species per hectare (Table 2). While the number of species per hectare is not a fair measure of biodiversity per se as the species-area curve is not linear [60], it is useful to assess the effectiveness of a given protected area. Clearly, Nectandra is highly effective at protecting a large number of pteridophyte species given its area. Furthermore, the collection curve indicates that additional, unsampled species may be present (Fig 2), adding to the value of this conservation area. A recent survey of the bryophytes of Nectandra found a similar number of species (188 spp.) and also suggested unsampled species remained due to the shape of the collection curve [21].

One reason for the high species richness at Nectandra may be its elevation (1,000 m to 1,200 m). Species richness of pteridophytes in the tropics generally reaches a maximum at mid-elevations on mountains, which is thought to be due to a combination of high humidity and moderate temperature [61]. Plot-based surveys of pteridophytes along elevational gradients in Costa Rica spanning *c*. 100 m to 3,000 m have found maximum richness at 1,000 m to 1,200 m [62,63]. Another possible explanation is the presence of secondary forest and reserve edges, which may contribute additional species that would otherwise not occur in primary forest at this elevation (i.e., the "edge effect" [64]). Nectandra has a high edge to area ratio due to its small size, and edge effects have been demonstrated for pteridophytes in Mexican montane forests [65]. While we did not collect data to specifically test this hypothesis, the number of

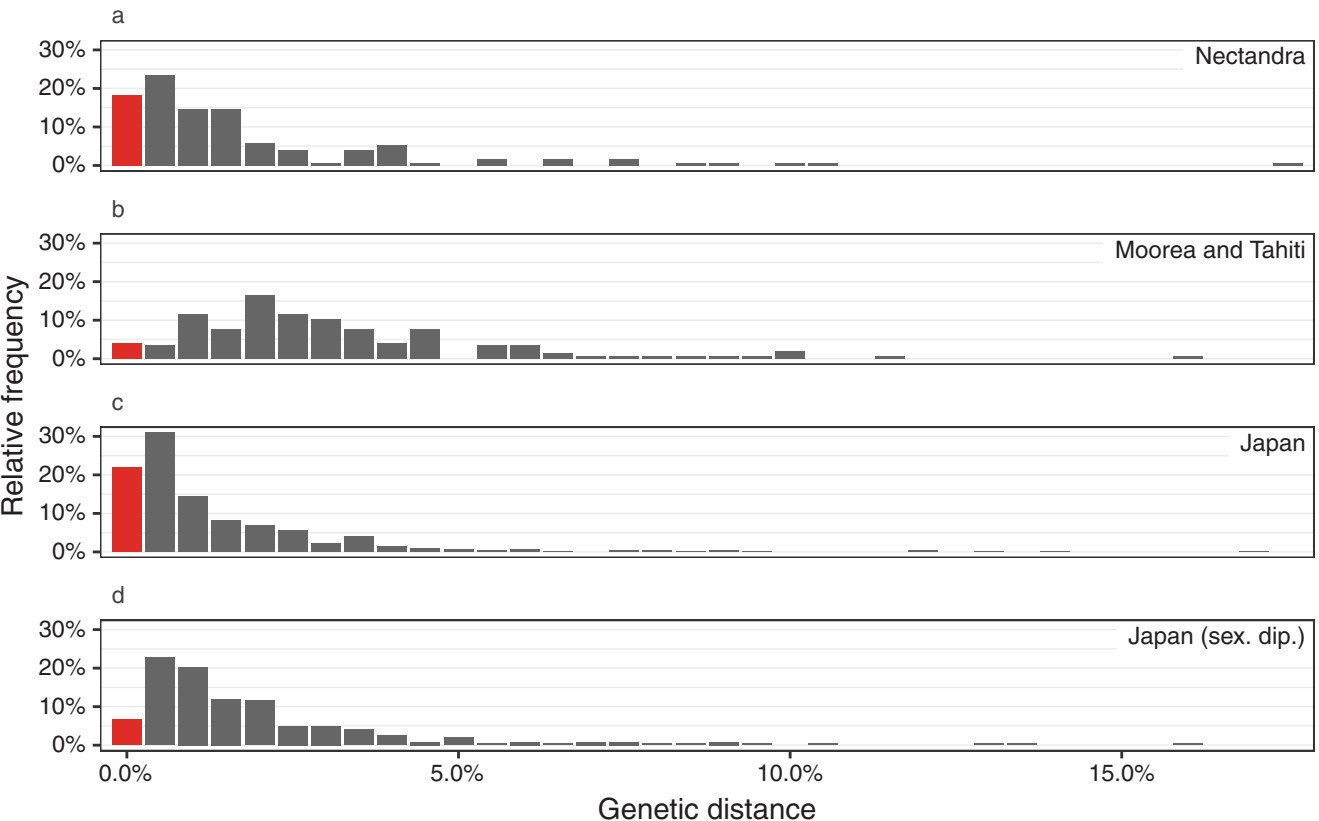

**Fig 3. Minimum interspecific *rbcL* distances for selected pteridophyte floras.** (a) Nectandra, (b) Moorea and Tahiti, (c) Japan (all species), and (d) Japan (sexual diploids only). Red bar indicates interspecific distance of zero, i.e., species that cannot be distinguished using *rbcL*.

taxa restricted to edges at Nectandra is probably no greater than *c.* 10 spp. (J. Nitta, pers. obs.), and only one non-native species (*M. torresiana*) was observed occupying disturbed areas. Furthermore, the high number of epiphytes (the most common growth form; Table 1) indicates that high humidity may support large numbers of species at Nectandra. Taken together, these observations suggest effect of elevation is likely more important than edge effects.

## Unidentified taxa and putative species complexes

Of the unidentified taxa, two are likely hybrids between distinct species. *Polyphlebium* sp1 (*Nitta 123* and *Nitta 2378*) shares very similar *rbcL* sequences with *Polyphlebium capillaceum* (L.) Ebihara & Dubuisson but differs from this species by having expanded laminae (vs. laminae reduced to a few cells on either side of the veins) and growing on rocks in stream beds (vs. growing epiphytically on tree ferns). *Polyphlebium* sp1 has 32 spores per sporangium, a condition that often indicates asexual reproduction via apogamy [66]. This may allow it to reproduce by spores despite not being able to complete normal meiosis. To our knowledge, a count of 32 spores per sporangium has not been previously reported from *Polyphlebium*, and more detailed study is needed to confirm the reproductive mode of this taxon. *Campyloneurum* sp1 (*Nitta 2308*) matches in *rbcL* exactly with *Campyloneurum angustifolium* (Sw.) Fée, but differs in morphology from *C. angustifolium* at Nectandra (*Nitta 782*) by having wider fronds (*c.* 1.5 cm. vs. *c.* 0.75 cm in *C. angustifolium*) and more rows of sori between the costa and the margin (3–4 vs. 1–2 in *C. angustifolium*). Clear, misshapen spores were observed in *Campyloneurum*

sp1, whereas *C. angustifolium* is a sexual diploid (2*n* = 72) with oblong spores [67]. The chloroplast is generally maternally inherited in ferns [68–70]. Therefore, it is likely that each of these taxa is a hybrid between the species with which they share *rbcL* as the mother and another unknown species as the father.

*Megalastrum* sp1 (*Nitta 727*) shares common aspects of morphology with *M. longipilosum*, but the *rbcL* sequences of *M. apicale*, *M. atrogriseum*, and *M. longipilosum* are extremely similar. The low divergence in *rbcL* and non-monophyly of some taxa (*M. apicale*) are consistent with the status of this group as a species complex, comprising several closely related diploid taxa and their hybrids; alternatively, *rbcL* alone may be too slowly evolving to distinguish recently diverged taxa in this group. Similarly, *Diplazium* also showed a high degree of morphological diversity but extremely low divergence in *rbcL* sequences, and at least one species, *D. carnosum*, appears to be non-monophyletic. Further study, in particular with regards to ploidal level and reproductive mode, is needed to determine the status of *Megalastrum* and *Diplazium* at Nectandra as possible species complexes. Also, sequencing of nuclear genes and more variable plastid markers is needed to clarify relationships for the putative hybrids (*Polyphlebium* sp1 and *Campyloneurum* sp1) as well as these putative species complexes.

## Molecular diversity and DNA barcode suitability

Unlike animals, there is no single DNA barcode available for plants that works to reliably identify species across all taxonomic groups [71]. *rbcL* has relatively high phylogenetic informativeness in ferns and lycophytes at the species level and is one of the most frequently sequenced plastid genes in molecular systematic studies of pteridophytes. In previous comparisons with other common plant barcode markers in pteridophytes, *rbcL* performs better than *trnH-psbA* and comparably with *matK* [6,38,39]. However, universal primers for *matK* in pteridophytes are lacking [38]. *rbcL* therefore is a reasonable choice for a single barcode marker in pteridophytes. However, there have been relatively few studies comparing the performance of *rbcL* as a barcode marker across different pteridophyte floras.

We find that the pteridophytes of Nectandra have a species identification failure rate intermediate between that of Moorea and Tahiti (French Polynesia) and Japan (Fig 3). The distribution of minimum interspecific distances may reflect the biogeographic history of each region. Costa Rica and Japan are both mainland areas (or formerly connected with the mainland in the case of Japan), whereas the islands of French Polynesia are extremely isolated and have never been connected to a continent. It is likely that more species in French Polynesia are recent immigrants that have mostly evolved elsewhere, compared to more species that have evolved *in situ* in Costa Rica and Japan. This would result in the observed distribution of high interspecific divergences for species from French Polynesia and low interspecific divergences for species in Costa Rica and Japan.

Furthermore, the high failure rate in Japan is likely due to a high rate of apogamy combined with the taxonomic practice of splitting apogamous and sexual forms into separate species [49]. When only sexual diploids are included, the failure rate in Japan drops to 7% and the distribution of minimum interspecific distances more closely resembles that of Nectandra (Fig 3).

The patterns identified here show that DNA barcoding strategies in different areas may need to take into account the biogeographical history of the study site. Whereas *rbcL* alone may be largely sufficient in oceanic islands like French Polynesia, a second, more variable barcode marker such as *trnLF* [72] or *matK* [38] is probably needed to improve species identification rates of recently diverged taxa in continental areas such as Japan or Costa Rica. The second marker need not have universal primers, since *rbcL* will indicate genus or family, and primers for the second marker can then be selected appropriately.

## Conclusions

The Nectandra Cloud Forest Reserve is clearly an important site for biodiversity of ferns and lycophytes in Costa Rica, harboring a high number of species for its relatively small area. Our surveys have already revealed multiple taxa that appear to be new to science, and more likely remain to be discovered. We demonstrated that *rbcL* can be used to reliably distinguish species in *c*. 82% of cases at Nectandra. This will enable future studies to investigate the ecology of neotropical pteridophytes in more depth, particularly with regards to the gametophytic phase. Other studies using *rbcL* as a DNA barcode to identify fern gametophytes to species have revealed particular clades and morphologies that tend to occur long distances from conspecific sporophytes [6,7], but such patterns have yet to be observed in the Neotropics. We hope our study will lead to more molecular ecological research incorporating field observations of gametophytes and contribute to the systematics of neotropical pteridophytes.

## Supporting information

**S1 Fig. Maximum-likelihood phylogenetic tree of ferns and lycophytes at the Nectandra Cloud Forest Reserve, Costa Rica inferred using IQ-TREE with 1,000 SH-like approximate likelihood ratio test (SH-aLRT) and ultra-fast bootstrap (UFboot) replicates each.** Tree rooted on lycophytes. Numbers at nodes indicate SH-aLRT support (%)/UFboot support (%); values less than 50% shown with "-"; values of 100% shown with "*"; completely blank nodes indicate identical sequences. For phylogram on right side, scale bar shows expected number of changes per site. Numbers after species name are J. H. Nitta specimen collection numbers except for *Abrodictyum rigidum* (*J.-Y. Dubuisson HV 1997–3*, Venezuela), *Trichomanes polypodioides* (*M. Kessler 8808*, Bolivia), and *Radiovittaria remota* (*R. Moran 3180a*, Costa Rica), which could not be sequenced successfully, so GenBank sequences were used instead (accessions AY095108, AY175795, and U21289, respectively). All J. H. Nitta specimens from Nectandra.
(PDF)

**S2 Fig. Maximum-likelihood phylogenetic tree of the family Cyatheaceae including all available *rbcL* sequences on GenBank and newly sequenced taxa from the Nectandra Cloud Forest Reserve, Costa Rica inferred using FastTree.** Tree rooted on species from Nectandra in family Dicksoniaceae. Numbers at nodes indicate local support values computed with the Shimodaira-Hasegawa test; values less than 50% not shown. For phylogram on right side, scale bar shows expected number of changes per site. Numbers after species names are GenBank accession numbers for sequences downloaded from GenBank or J. H. Nitta specimen collection numbers for sequences newly obtained by this study. Newly obtained ingroup sequences highlighted in yellow.
(PDF)

**S3 Fig. Maximum-likelihood phylogenetic tree of grammitid ferns (subfamily Grammitidoideae) including all available *rbcL* sequences on GenBank and newly sequenced taxa from the Nectandra Cloud Forest Reserve, Costa Rica inferred using FastTree.** Tree rooted on species from Nectandra in subfamily Polypodioideae. Numbers at nodes indicate local support values computed with the Shimodaira-Hasegawa test; values less than 50% not shown. For phylogram on right side, scale bar shows expected number of changes per site. Numbers after species names are GenBank accession numbers for sequences downloaded from GenBank or J. H. Nitta specimen collection numbers for sequences newly obtained by this study. Newly obtained ingroup sequences highlighted in yellow.
(PDF)

**S1 Table. GenBank accession numbers of sequences analyzed in this study.** All specimens from the Nectandra Cloud Forest Reserve, Alajuela Province, Costa Rica, except for *Abrodictyum rigidum* (Sw.) Ebihara & Dubuisson (*J.-Y. Dubuisson HV 1997–3*, Venezuela), *Radiovittaria remota* (Fée) E.H. Crane (*R. Moran 3180a*, Costa Rica), and *Trichomanes polypodiodes* L. (*M. Kessler 8808*, Bolivia). Accessions beginning with 'MW' were newly generated by this study.
(CSV)

**S2 Table. A checklist of ferns and lycophytes observed at the Nectandra Cloud Forest Reserve, Costa Rica.** All voucher specimens deposited in UC with duplicates when available at CR, GH, and TI. For additional data on each voucher specimen, see Dryad repository [51].
(CSV)

# Acknowledgments

We are grateful to Evelyne and David Lennette for allowing us to study the pteridophytes of Nectandra over multiple field seasons and for providing logistical support. Freddy Castillo Arroyo assisted with fieldwork. Paulo Labiak contributed to the identification of *Lellingeria hombersleyi*. Paulo Labiak and Michael Sundue reviewed the manuscript. This study is dedicated in memory of Álvaro Ulgade, father of the Costa Rican national park system, co-founder of Nectandra, and an inspiration to all those who seek to understand and protect biodiversity.

# Author Contributions

**Conceptualization:** Joel H. Nitta, Alan R. Smith.

**Data curation:** Joel H. Nitta.

**Formal analysis:** Joel H. Nitta, Alan R. Smith.

**Funding acquisition:** Atsushi Ebihara.

**Investigation:** Joel H. Nitta.

**Methodology:** Joel H. Nitta, Alan R. Smith.

**Resources:** Atsushi Ebihara.

**Visualization:** Joel H. Nitta.

**Writing – original draft:** Joel H. Nitta.

**Writing – review & editing:** Joel H. Nitta, Atsushi Ebihara, Alan R. Smith.

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
