## [Decision Letter · Decision Letter 0]

16 Jul 2020

PONE-D-20-11593

A taxonomic and molecular survey of the pteridophytes of the Nectandra Cloud Forest Reserve, Costa Rica

PLOS ONE

Dear Dr. Nitta,

Thank you for submitting your manuscript to PLOS ONE. After careful consideration, we feel that it has merit but does not fully meet PLOS ONE’s publication criteria as it currently stands. Therefore, we invite you to submit a revised version of the manuscript that addresses the points raised during the review process.

Please, consider  especially  rev.#1's comments on improving bibliographic references and the taxonomic survey. Further details on Material & Methods are also welcome.

We look forward to receiving your revised manuscript.

Kind regards,

Paulo Takeo Sano, Ph.D.

Academic Editor

PLOS ONE

Journal Requirements:

3.We note that [Figure(s) 1] in your submission contain [map/satellite] images which may be copyrighted. All PLOS content is published under the Creative Commons Attribution License (CC BY 4.0), which means that the manuscript, images, and Supporting Information files will be freely available online, and any third party is permitted to access, download, copy, distribute, and use these materials in any way, even commercially, with proper attribution. For these reasons, we cannot publish previously copyrighted maps or satellite images created using proprietary data, such as Google software (Google Maps, Street View, and Earth). For more information, see our copyright guidelines: http://journals.plos.org/plosone/s/licenses-and-copyright.

1.    You may seek permission from the original copyright holder of Figure(s) [1] to publish the content specifically under the CC BY 4.0 license. 

Additional Editor Comments (if provided):

Please consider all comments from both reviewers, especially rev.#1's comments on improving bibliographic references and the taxonomic survey. Rev.#1 also asked for further clarification on some methods.

Reviewers' comments:

Reviewer's Responses to Questions

**Comments to the Author**

1. Is the manuscript technically sound, and do the data support the conclusions?

Reviewer #1: Yes

Reviewer #2: Yes

2. Has the statistical analysis been performed appropriately and rigorously? 

Reviewer #1: Yes

Reviewer #2: Yes

3. Have the authors made all data underlying the findings in their manuscript fully available?

Reviewer #1: Yes

Reviewer #2: Yes

4. Is the manuscript presented in an intelligible fashion and written in standard English?

Reviewer #1: Yes

Reviewer #2: Yes

5. Review Comments to the Author

Reviewer #1: Most of my suggestions were provided in the annotated file provided. Some major comments are below:

1- I think many relevant citations are missing for the Taxonomic part of the manuscript. Besides the general floras published by Lellinger and Moran et al., I strongly recommend the authors to consult, and cite, the relevant monographs (taxonomic revisions) that are available for many of the taxa, as well as recent taxonomic works on Costa Rican ferns. I think it is important to acknowledge the contribution of these studies to the understanding of biodiversity.

2- Line 102. Please, list the taxa that you considered the most difficult to identify.

3- Line 105. I think it would be nice to explain why rbcL was chosen for this study in the Mat & Meth Section.

4- Please, include something in the Mat & Meth about model calculations. By the way, why you used BIC for the ML analysis? Why not to use AIC?

5- Taxonomic survey: Some improvements could be made to this part. For instance, mention how many are ferns and how many are lycophytes, and what is the amount of epiphytes, terrestrial, epipetric, etc.. Also, are there exotic/invasive taxa that might be contributing to the diversity of Nectandra? Were those included in your analysis?

6- Barcode analysis: This is why rbcL may not be the best to infer species boundaries, and it may not be enough to state that there are "species complexes" based on the phylogenetic results. (Even though I think this might be true in some cases...). Please, consider clarifying this paragraph.

7- About what would explain the richness in Nectandra, what about humidity? Because this region is located at the Atlantic slopes, it has a high humidity along the year, which contributes to the occurrence of many epiphytes. Because it has been noted that epiphytes represent a considerable part of the diversity in some areas, I wonder whether this is not the case also in Nectandra. By the way, this is another reason it would be interesting to mention how many species are epiphytes, terrestrial, epipetric, etc.

8- As for the comparison provided for Tahiti, Japan and Nectandra’s floras, I think another possibility is the isolation of Tahiti, which would preclude recurrent events of dispersal from other areas, keeping the isolation of local populations. In both CR and Japan, there are more possibilities of gene flow between other areas, adding to the complexity of local populations.

Reviewer #2: The authors present a checklist to a reserve that is accomplished through a combination of field work, herbarium study, and DNA barcoding using the chloroplast rbcL marker. The work appears to have been conducted with the utmost care. It is clearly presented and well written. It will act as a gold standard by which similar projects will be compared. I have made some additional comments, but I have a hard time finding errors in this work I congratulate the authors on a job well done.

Signed,

Michael Sundue

I am happy to see that this work adheres to open and reproducible science. The effort and thoroughness of providing the fully reproducible manuscript using Docker and Drake are appreciated. Nonetheless, I was unable to reproduce the manuscript. The address https://github.com/joelnitta/nectandra_ferns

returned a 404 page not found error and attempting docker pull joelnitta/nectandra_ferns

returned a ‘manifest unknown error’. Perhaps I am missing something simple.

195 change “was in generally good agreement” to “generally in agreement”

The evidence in favor of the new taxa is clear and the arguments used to explain them is logical.

While I appreciate the brevity of the paper. Personally, I would like to see more discussion of the potentially novel taxa with reference to the current state of the taxonomy in each group. That said, the authors are perfectly justified in maintaining their brief discussion if they prefer.

The trees presented in the supplement show species of tree ferns that are highlighted in yellow some of which are non-monophyletic. Yet there is no discussion of these nor any other tree fern for that matter. Why were these results excluded from the discussion? The same is true for some polypodiaceae.

Your Mycopteris taxifolia may be M. costaricense. See Sundue 2014 “Mycopteris, a new neotropical genus of grammitid ferns”

6. PLOS authors have the option to publish the peer review history of their article (what does this mean?). If published, this will include your full peer review and any attached files.

Reviewer #1: **Yes: **Paulo Labiak

Reviewer #2: **Yes: **Michael Sundue

---

## [Author Response · Author response to Decision Letter 0]

28 Sep 2020

2020-09-29

Dear Dr. Paulo Takeo Sano,

Thank you very much for handling our MS. We are happy for the chance to respond to the reviewers. Please see our responses below (all line numbers in our responses refer to the revised MS unless otherwise indicated).

There is also a summary of other changes we made to the MS besides those requested by the reviewers following our responses.

Best Regards,

Joel Nitta

Joel H. Nitta, Ph.D.

Project Research Associate

Iwasaki Lab

Department of Biological Sciences

The University of Tokyo

nitta@bs.s.u-tokyo.ac.jp

https://www.joelnitta.comhttp://iwasakilab.bs.s.u-tokyo.ac.jp/

---

Reviewer #1: Most of my suggestions were provided in the annotated file provided. Some major comments are below:

1- I think many relevant citations are missing for the Taxonomic part of the manuscript. Besides the general floras published by Lellinger and Moran et al., I strongly recommend the authors to consult, and cite, the relevant monographs (taxonomic revisions) that are available for many of the taxa, as well as recent taxonomic works on Costa Rican ferns. I think it is important to acknowledge the contribution of these studies to the understanding of biodiversity.

RESPONSE: We have added one more flora and seven recent monographs (line 102).

2- Line 102. Please, list the taxa that you considered the most difficult to identify.

RESPONSE: We have added examples of taxa that were difficult to identify (Megalastrum and Didymoglossum, line 110).

3- Line 105. I think it would be nice to explain why rbcL was chosen for this study in the Mat & Meth Section.

RESPONSE: We have added our reasons for selecting rbcL (lines 111-113).

4- Please, include something in the Mat & Meth about model calculations. By the way, why you used BIC for the ML analysis? Why not to use AIC?

RESPONSE: We have added more details about the procedure used for model selection (lines 123-127). We used BIC because it is the default in IQ-tree. Practically speaking, in our analysis, the same model was selected by AIC, AICc, and BIC (GTR+I+G4), so it makes no difference which one was used. We now mention this in the results (line 216).

5- Taxonomic survey: Some improvements could be made to this part. For instance, mention how many are ferns and how many are lycophytes, and what is the amount of epiphytes, terrestrial, epipetric, etc.. Also, are there exotic/invasive taxa that might be contributing to the diversity of Nectandra? Were those included in your analysis?

RESPONSE: We have added a count of the number of ferns vs. lycophytes (line 173). We have added a description in the text of the number of taxa observed with different growth habits (lines 173-174), as well as a new table summarizing this information (new Table 1). There is only one non-native species (Macrothelypteris torresiana), which we now mention. We also now exclude the non-native species from the comparison of richness across sites and the DNA barcode analysis.

6- Barcode analysis: This is why rbcL may not be the best to infer species boundaries, and it may not be enough to state that there are "species complexes" based on the phylogenetic results. (Even though I think this might be true in some cases...). Please, consider clarifying this paragraph.

RESPONSE: Thanks to the reviewer for pointing out this possibility. While it is not clear which paragraph the reviewer is referring to, we have clarified the text in several areas to reflect that rbcL alone may not be sufficient for species discrimination at Nectandra, as follows:

- We removed "indicative of species complexes" (line 216, original MS) from the Results.

- We now say "consistent with the status of this group as a species complex" (instead of "indicative") in the Discussion (line 309).

- We now state "alternatively, rbcL alone may be too slowly evolving to distinguish recently diverged taxa" in the Discussion (lines 310-311).

- We added a paragraph to the Discussion about the need for a second marker in continental areas like Costa Rica (lines 347-354).

7- About what would explain the richness in Nectandra, what about humidity? Because this region is located at the Atlantic slopes, it has a high humidity along the year, which contributes to the occurrence of many epiphytes. Because it has been noted that epiphytes represent a considerable part of the diversity in some areas, I wonder whether this is not the case also in Nectandra. By the way, this is another reason it would be interesting to mention how many species are epiphytes, terrestrial, epipetric, etc.

RESPONSE: Thanks to the reviewer for pointing this out. We have now added mention of humidity to the Discussion, including the observation of high numbers of epiphytes at Nectandra (lines 283-284). We also added a table summarizing growth habits (new Table 1).

8- As for the comparison provided for Tahiti, Japan and Nectandra’s floras, I think another possibility is the isolation of Tahiti, which would preclude recurrent events of dispersal from other areas, keeping the isolation of local populations. In both CR and Japan, there are more possibilities of gene flow between other areas, adding to the complexity of local populations.

RESPONSE: We already explicitly state "the islands of French Polynesia are extremely isolated" in the sentence directly preceding the commented text. We do not see any need to revise the text further.

---

Following are our responses to comments from Reviewer 1 included as annotations in the PDF. All line numbers for the reviewer's comments refer to the original MS.

Lines 69-70: Why, specifically, the herpetofauna and bryophytes were mentioned here? are these the only surveys carried out in the reserve? If so, please, include this information.

RESPONSE: Yes; these are the only published surveys previously carried out at the reserve. Providing information about the biodiversity of the site for other organisms is useful background information. We now mention that these are previous biological surveys carried out at Nectandra (line 69). However, saying they are the "only" previous studies seems superfluous. We believe the reader would understand that we are citing all relevant literature and haven't left out anything on purpose.

Line 97: Haven't you asked any experts for help with some particular groups? Also, I think it is important to cite the relevant monographs and recent taxonomic works on Costa Rican ferns. 

RESPONSE: Yes! Indeed, Paulo Labiak answered a query about Lellingeria hombersleyi while we were preparing the manuscript (in case there is any question, this communication took place prior to submission, was strictly about this specimen only, and did not include any other information about this manuscript). We apologize for forgetting to mention it in the original MS. We have added consulting taxonomic experts to the methods (lines 103-104) and mention any such contributions in the Acknowledgments (lines 371-373). We have also added additional references used for identification (see response to Reviewer 1, Point 1).

Line 102: Please, list the taxa that you considered the most difficult to identify.

RESPONSE: See response to Reviewer 1, Point 2.

Line 103-105 (with regards to specimens from other areas in Costa Rica): Is this listed somewhere? I think it is important to specify which ones are not from Nectandra.

RESPONSE: We have removed these specimens from analysis (see “additional corrections” at the end of this document).

Line 114 (with regards to phylogenetic analysis): No model was estimated?

RESPONSE: More details have been added (see response to Reviewer 1, Point 4).

Line 159: Please, specify how many are ferns and how many are lycophytes.

RESPONSE: This has been added (see response to Reviewer 1, Point 5).

Line 187 (with regards to difficulty in sequencing some samples): What about primer specificity?

RESPONSE: We now mention this possibility (line 209).

Line 193-194 (with regards to model of DNA evolution): This needs to be mentioned in the Mat& Met section. What program was used? Also, I think it would be more appropriated to use AIC for ML analysis, instead of BIC.

RESPONSE: We disagree that the particular model used must be reported in the materials and methods, and not results. The outcome of model selection is a result of an algorithm (as well as used in phylogenetic analysis). Furthermore, more details about the model selection procedure have been added to the methods (see response to Reviewer 1, Point 4).

Line 216 (with regards to putative species complexes): Couldn't it be due to the lack of resolution based on the rbcL sequences instead? rbcL is a very conservative gene, and perhaps not the best one to be used at the species level.

RESPONSE: See response to Reviewer 1, Point 6.

Lines 225-226 (with regards to statement that 18% of species at Nectandra share identical rbcL sequences with at least one other taxon): See, this is why rbcL may not be the best to infer species boundaries, and it may not be enough to state that there are "species complexes" based on the phylogenetic results. (Even though I think this might be true in some cases...)

RESPONSE: See response to Reviewer 1, Point 6.

Lines 253-254 (with regards to additional species in secondary forest and reserve edges): Are there non-native species recorded for these disturbed areas? Could they be contributing to the diversity in Nectandra? I think it is worth including a sentence about it.

RESPONSE: There is only one non-native species (Macrothelypteris torresiana). We now mention this in the discussion of diversity at Nectandra (line 282).

Line 259 (with regards to the effect of elevation on number of species): What about humidity? Because this region is located at the Atlantic slopes, it has a high huidity along the yearm which contributes to the occurence of many epiphytes. Because it has been noted that epiphytes represent a considerable part of the diversity in some areas, I wonder whether this is not the case also in Nectandra?

By the way, I think it would be interesting to mention how many species are epiphytes, terrestrial, epipetric, etc...

RESPONSE: See response to Reviewer 1, Point 7.

Line 271 (with regards to Campyloneurum sp1): Alexander ARojas, a Costa Rican pteridologist, has recently published some new species in this complex. Perhaps your specimen could be one of them?

RESPONSE: We checked the species in Rojas 2017 (Am. J. Plant Sci.) and confirmed that our specimen is not one of these. We also added Rojas 2017 (Ref. 35) to the list of monographs that we consulted for species identification (line 102).

Lines 290-291, 296-297 (with regards to use of rbcL in molecular systematics in ferns and as a barcode marker): These two sentences are kind of contradictory. Please, rephrase.

RESPONSE: We disagree with the reviewer on this point. The first sentence establishes that rbcL is widely used in molecular systematic studies of pteridophytes; the second says that there have been relatively few studies comparing the performance of rbcL as a barcode marker across different pteridophyte floras. We see no contradiction in these statements; systematic studies and barcode studies are two different things.

Lines 304-306: I think that another possibility is the isolation of Tahiti. This would preclude recurrent events of dispersal from other areas, keeping the isolation of local populations. In both CR and Japan, there are more possibilities of gene flow between other areas, adding to the complexity of local populations.

RESPONSE: See response to Reviewer 1, Point 8.

Lines 362-364: Authors?

RESPONSE: We have corrected this reference (line 405), as well as others that were formatted incorrectly in the last version.

---

Reviewer #2: The authors present a checklist to a reserve that is accomplished through a combination of field work, herbarium study, and DNA barcoding using the chloroplast rbcL marker. The work appears to have been conducted with the utmost care. It is clearly presented and well written. It will act as a gold standard by which similar projects will be compared. I have made some additional comments, but I have a hard time finding errors in this work I congratulate the authors on a job well done.

RESPONSE: We thank the reviewer for the encouraging comments. Indeed, we hope this method will be applied in other areas and contribute to our understanding of biodiversity.

I am happy to see that this work adheres to open and reproducible science. The effort and thoroughness of providing the fully reproducible manuscript using Docker and Drake are appreciated. Nonetheless, I was unable to reproduce the manuscript. The address https://github.com/joelnitta/nectandra_ferns returned a 404 page not found error and attempting docker pull joelnitta/nectandra_ferns returned a ‘manifest unknown error’. Perhaps I am missing something simple.

RESPONSE: The github repository is still set to private, so that is why the attempting to open the URL resulted in a 404 error. The github repository will be made public upon acceptance. We knew this would happen, so that is why we made the code available via a dropbox link for reviewers to use in the meantime (line 154, original manuscript). Apparently the reviewer did not notice this. We apologize for any confusion. We corrected the docker image settings: "docker pull joelnitta/nectandra_ferns" should now work properly. 

195 change “was in generally good agreement” to “generally in agreement”

RESPONSE: Done (line 217, revised MS).

The evidence in favor of the new taxa is clear and the arguments used to explain them is logical. While I appreciate the brevity of the paper. Personally, I would like to see more discussion of the potentially novel taxa with reference to the current state of the taxonomy in each group. That said, the authors are perfectly justified in maintaining their brief discussion if they prefer.

RESPONSE: Unfortunately, it was beyond the scope of this paper to do a thorough investigation of the potentially novel taxa. We therefore choose to keep our discussion of these to a minimum and suggest directions for further study.

The trees presented in the supplement show species of tree ferns that are highlighted in yellow some of which are non-monophyletic. Yet there is no discussion of these nor any other tree fern for that matter. Why were these results excluded from the discussion? The same is true for some polypodiaceae.

RESPONSE: We believe this is already clear in the original text: The purpose of the trees in Figs S2 and S3 is to check the placement of genera that were recovered as non-monophyletic in the sampling of species only at Nectandra with a broader sampling, as described in the methods (lines 129-133). We state in the Results that these are recovered in their expected genera in the broadly sampled trees (lines 232-236), and explain the yellow highlighting in the caption for the supplemental figures (lines 608-609, 618-619). There is no need to discuss these trees further in the Discussion, since they do not pertain directly to analysis at Nectandra, and have been presented in the Results.

Your Mycopteris taxifolia may be M. costaricense. See Sundue 2014 “Mycopteris, a new neotropical genus of grammitid ferns”

RESPONSE: Thanks to the reviewer for pointing this out. Upon closely re-examining species originally identified as Mycopteris taxifolia, we realized that the reviewer is correct, and that these are indeed Mycopteris costaricensis ("costaricense" in the reviewer's comment is a spelling error). We have updated our data and the manuscript appropriately.

---

We made the following additional corrections to the manuscript (all line numbers refer to the revised manuscript):

Line 4: Correct affiliation (from "University of Tokyo" to "The University of Tokyo")

Line 23: Change the number of previously undescribed taxa that appear to be of hybrid origin from three to two.

Lines 77-79: Add references for source of elevation data and software used to create map.

Line 80: Correct date of photo from 1995 to 1992.

Lines 83-86: Add CC BY license for photos.

Line 97: Correct "TNS" to "TI".

Line 113, others subsequently: In the original submission, we included new sequences of two Costa Rican specimens collected from areas outside of the Nectandra reserve. We realized that this was not covered by our molecular permits (R-CM-RN-001-2014-OT-CONAGEBIO and R-CM-RN-002-2017-OT-CONAGEBIO), which were specifically for specimens from Nectandra. We have removed these sequences from the analyses, removed all references to them from the MS, and will not be submitting them to GenBank. This has no impact on the main findings of the paper. To see other places where edits were made to the MS regarding these specimens, please see “nectandra_ferns_changes_2020-04-22_to_2020-09-29.docx".

Line 134: Add one more permit (R-CM-RN-002-2017-OT-CONAGEBIO).

Line 287: Change heading to from "Unidentified taxa and species complexes" to "Unidentified taxa and putative species complexes"

Lines 297-301: Clarify differences in morphology between Campyloneurum sp1 and Campyloneurum angustifolium by mentioning the number of rows of sori between the costa and margin, and adding measurements for lamina width.

References: Fix several references.

---

## [Editor Report · Decision Letter 1]

12 Oct 2020

A taxonomic and molecular survey of the pteridophytes of the Nectandra Cloud Forest Reserve, Costa Rica

PONE-D-20-11593R1

Dear Dr. Nitta,

We’re pleased to inform you that your manuscript has been judged scientifically suitable for publication and will be formally accepted for publication once it meets all outstanding technical requirements.

Kind regards,

Paulo Takeo Sano, Ph.D.

Academic Editor

PLOS ONE

Additional Editor Comments (optional):

The text improved a lot after the changes made. Thank you for the informations. I wish you success with this paper.
---

## [Editor Report · Acceptance letter]

10 Nov 2020

PONE-D-20-11593R1 

A taxonomic and molecular survey of the pteridophytes of the Nectandra Cloud Forest Reserve, Costa Rica 

Dear Dr. Nitta:

I'm pleased to inform you that your manuscript has been deemed suitable for publication in PLOS ONE. Congratulations! Your manuscript is now with our production department. 

Kind regards, 

on behalf of

Dr. Paulo Takeo Sano 

Academic Editor

PLOS ONE